# Synthesizing Stakeholders Perspectives on Online Psychological Interventions to Improve the Mental Health of the Italian Population during the COVID-19 Pandemic: An Online Survey Study

**DOI:** 10.3390/ijerph19127008

**Published:** 2022-06-08

**Authors:** Vanessa Bertuzzi, Michelle Semonella, Gianluca Castelnuovo, Gerhard Andersson, Giada Pietrabissa

**Affiliations:** 1Department of Psychology, Catholic University of Milan, 20123 Milan, Italy; vanessa.bertuzzi@unicatt.it (V.B.); gianluca.castelnuovo@unicatt.it (G.C.); 2Department of Psychology, Bar-Ilan University, Ramat-Gan 52900, Israel; michelle.semonella@biu.ac.il; 3Psychology Research Laboratory, Istituto Auxologico Italiano IRCCS, 20123 Milan, Italy; 4Department of Behavioural Science and Learning, Linköping University, 58183 Linköping, Sweden; gerhard.andersson@liu.se; 5Department of Clinical Neuroscience, Karolinska Institute, 17177 Stockholm, Sweden

**Keywords:** COVID-19, online psychological intervention, stakeholders survey, clinical psychology

## Abstract

The COVID-19 pandemic is expected to significantly increase the prevalence of mental health problems, thus raising the need for psychological support interventions around the world. Online psychological interventions have already been shown to be an effective solution to promote psychological treatments. Nevertheless, planning and developing an online intervention, involving possible stakeholders, might facilitate the dissemination of, willingness to use, and success of the future intervention. This study aims to explore and compare the experiences that Italians living in Italy and abroad had with available support services during the COVID-19 pandemic, their needs, and attitudes, as well as possible barriers to online psychological interventions. A sample of 1024 Italians (F = 69.8%; mean age = 41.3; SD = 15.3) was recruited through social media platforms and personal contacts and they were asked to complete an online survey. Results showed that perceived psychological distress during the COVID-19 pandemic improved. In Europe, psychological support was delivered mainly in person (69.0%), while online interventions were primarily used in extra-European countries (57%). Then, only 44% of the total sample was interested in trying an online psychological intervention. Various advantages and disadvantages were defined by stakeholders: The main advantages were the reduction in geographical distances, economic reasons, and the reduction in the waiting list; The main disadvantages were problems with technology, low motivation of users, and privacy/safety reasons. These data made it possible to improve the knowledge regarding the views and attitudes that Italians have about online psychological interventions, and shed light on how to increase the uptake of digital health.

## 1. Introduction

The coronavirus (COVID-19) pandemic has been having a great impact on individuals’ mental health. Not only did the restrictive measures employed by governments around the world to reduce the spread of the virus, including social distancing, the use of sanitary masks, and mobility restrictions [1] drastically alter people’s lives [2,3] but uncertainties and fear associated with the virus outbreak, lack of clear information, and the economic losses resulting from social isolation [4] also had a strong, negative impact on people’s lives [5], leading to great distress and psychological problems [1,6,7].

Reactions to the pandemic comprise maladaptive behaviors (e.g., drug use, sedentary habits, unhealthy diet, insomnia) [8,9,10] emotional distress, and mood disorders (e.g., anxiety, depression) [11,12,13,14,15]. A study, aimed at measuring the impact of lockdown and the COVID-19 pandemic on the mental health of the general population in Italy, revealed a meaningful increase in depressive symptoms, anxiety, insomnia, and perceived stress [16].

However, this is not the first time that humanity has faced an epidemic. Over the last century, many pandemics such as the Spanish flu, severe acute respiratory syndrome (SARS), Middle East respiratory syndrome (MERS), Ebola, and swine flu have emerged, and the existing literature suggested that public health emergencies and related quarantine requirements represent a risk factor for the onset and the increased severity of psychopathological symptoms [17,18,19,20,21,22].

Still, during previous pandemics, social isolation and mobility restrictions prevented the delivery of psychological support interventions; however, nowadays, digital solutions can offer therapeutic approaches and support positive behavioral change on a large scale. They are accessible at any time and from anywhere, providing help on-demand to individuals living in remote and rural areas, as well as to those unable to attend in-person sessions due to health-related issues, reduced mobility, incarceration, and/or working off-shore. They are also convenient, easy to use, and anonymous [23].

With the advent of the COVID-19 pandemic, the potential of digital technologies in our daily lives [24], and their important role in complementing conventional public health measures unquestionably emerged [25].

Still, while digital technologies offer tools for supporting a pandemic response, they are not a silver bullet, and recognizing both barriers and facilitators for the uptake and dissemination of digital solutions has become urgent to promptly provide adequate support interventions to the population. Specifically, in the field of mental health, ensuring widespread and trustworthy digital access requires key stakeholders in the digital domain, such as the users themselves, to be long-term partners in the processes of planning, implementing, and evaluating online psychological support interventions.

It would help to create user-friendly programs with a higher likelihood of adoption and dissemination [26,27], besides helping to overcome much of the stigma that continues to be associated with mental health issues [28].

In this scenario, the present stakeholders’ survey aims to explore and compare the experiences that Italians living in Italy and abroad had with the available support services during the COVID-19 pandemic, their needs, and attitudes, as well as possible barriers to the use of online psychological interventions.

Results are going to support the implementation and dissemination of the RinasciMENTE program [29], an internet-based self-help intervention specifically designed to address psychological problems experienced by the Italian population during the outbreak.

## 2. Materials and Methods

### 2.1. Study Design

Data for the study were collected using an online survey, which was developed and administered using the tailored design method [30]. This technique was chosen because an online questionnaire can easily be disseminated across a large number of stakeholders. The survey was first piloted on a small group of participants (*n* = 10). Based on their feedback, minor adaptations were made, and a final version of the survey was created. A qualitative approach was applied to synthesize the results and form meta-inferences at the end of the data collection.

### 2.2. Ethical Statement

The study was approved by the Research Ethics Board of the Catholic University of Milan, Milan, Italy (ID: 25-21). All procedures performed in the study followed the ethical standards of the institutional and/or national research committee and the Declaration of Helsinki and its later amendments or comparable ethical standards.

### 2.3. Participants

A total of 1207 subjects took part in the survey. Among these, 183 respondents did not fully complete the survey and were excluded from the data analysis. Demographic information about the 1024 remaining respondents are described in the Appendix A.

Inclusion criteria for the participants in the study were: (A) being a native Italian speaker; (B) being 18 years old or older; (C) having self-reported basic computer knowledge and skills; and (D) providing online informed consent to participate in the study.

### 2.4. Sample Size

To determine whether the sample size was large enough, an a priori power analysis using SurveyMonkey was run as shown in previous studies [31]. Results showed that the minimum sample size required to detect a small-to-medium effect size, given the power of 0.95, a critical alpha of 0.05, was around 778 respondents. Based on this calculation, the size of our sample was deemed adequate.

### 2.5. Measures

The topic guides for the online survey were strongly oriented toward the research questions informed by the RE-AIM framework [32].

Thus, the topics addressed stakeholders’ (i) experiences with online psychological interventions for mental health problems, (ii) opinion regarding such interventions including adequacy, usefulness, and accessibility, (iii) knowledge and values regarding such interventions, (iv) point of view regarding potential advantages and disadvantages towards the use of online psychological interventions, and (v) attitudes towards the use of online psychological interventions.

Additionally, the development of the instrument was informed by the E-COMPARED online survey [33], which was used to explore stakeholders’ views on digital treatment for depression.

The survey included 31 questions concerning: demographic information (e.g., age, gender, country of residence, education level) (6 questions); current occupation and related change due to the pandemic (4 questions); ownership of a personal computer and stable internet connection at home (2 questions); the suffering from psychological distress before and during the COVID-19 pandemic (11 questions); and possible advantages and disadvantages of an online psychological intervention to support individuals’ mental health (7 questions).

Questions used a multiple-choice response format (mainly for demographic information) or a 5-point Likert scale, with Likert scales ranging from 0 = not at all useful to 5 = very useful (utility of internet-based interventions), or 0 = not at all appropriate to 5 = very appropriate (appropriateness of online psychological interventions), or 0 = very disadvantageous to 5 = very advantageous.

Moreover, two questions allowed the participants to provide multiple answers among 9 possible options (list of advantages/disadvantages of online therapy) [34]. Participants were asked to respond yes or no to all the remaining questions (*n* = 14) (e.g., information about COVID-19 infection, presence of psychological distress, etc.).

The full survey is available as a Appendix A.

### 2.6. Procedure

Following a criterion-based sampling strategy [35,36], participants were recruited among the general population via social media platforms, personal contacts, and on the RinasciMENTE project Facebook page, by applying a snowballing sampling method [37]. Moreover, the link to the survey was posted on social networking pages dedicated to Italians living abroad (e.g., “Italians in London”, “Italians in New York”, “Italians in Seoul”, etc.). Each participant entered the Qualtrics platform using the invitation link. They were first provided with information on the inclusion and exclusion criteria, as well as details about the study and the types of questions comprising the survey. Then, respondents were asked to provide their online informed consent to take part in the study by checking the box “I consent to the processing of personal data”, and they were successively directed to the online survey.

No personal data (e.g., name or e-mail address) were needed to access the survey. Data were transmitted via a secured connection and stored on a secured and password-protected server.

### 2.7. Data Analysis

Statistical analyses were performed using SPSS software ver. 24.0 [38].

Demographic characteristics were reported as means and standard deviations for continuous variables, and frequencies and percentages for categorical variables. Descriptive statistics and graphic representations were used to characterize the sample and correlations were made, where relevant to the study aim.

## 3. Results

Seven hundred and fifteen out of 1024 (69.8%) respondents were female. The mean age of the sample was 41.3 years (SD = 15.3; age range: 18-89), with a prevalence of young adults (*n* = 553, 54.2%). Most of the respondents had a degree (*n* = 432, 42.1%) and 49.8% lived in Italy (*n* = 510). The geographical distribution of the sample is reported in Figure 1.

Before the advent of the COVID-19 pandemic, 38.7% of the sample had a full-time job (*n* = 396), and with the advent of the COVID-19 pandemic, 76.5% of them maintained this job position (*n* = 156). Most of the sample (*n* = 876; 85.9%) did not contract COVID-19, while a loved one did in 54.1% (*n* = 552) of the respondents; furthermore, of the 144 subjects who were infected with the virus, as many as 66.7% (*n* = 96) experienced psychological distress and emotional difficulties, while only 33.3% (*n* = 48) did not suffer a psychological impact from the infection.

Moreover, 100 (69.4%) out of the 144 people who experienced COVID-19 infection also had a loved one infected.

Nearly half of the sample stated they had never experienced psychological distress before the pandemic (*n* = 557; 54.4%), while, with the advent of COVID-19, 63.6% (*n* = 651) of the respondents began to suffer from mental health problems. Data are reported in Figure 2.

Results also showed that 301 out of 624 subjects who experienced psychological distress had a symptoms duration of less than 1 year (*n* = 298; 48.0%), followed by those who experienced psychological distress from 1 to 5 years (*n* = 225; 35.9%), from 6 to 10 years (*n* = 40; 6.4%), and for more than 10 years (*n* = 61; 9.7%), respectively.

Still, only 16.9% (*n* = 172) of the total sample asked for psychological support, that, in most cases, was delivered face-to-face (*n* = 84; 53.9%).

Those who attended online meetings with a therapist were mainly Italians living abroad (*n* = 34, 48.6%), with a prevalence of extra-European countries (Canada, *n* = 4, 100%; Turkey, *n* = 4, 100%; South Korea, *n* = 4, 36.4%; Chile, *n* = 4, 100%; USA, *n* = 4, 100%; China, *n* = 3, 50%; India, *n* = 1, 50%; and South Africa, *n* = 1, 100%) compared to European countries (Austria, *n* = 5, 100%; and Spain, *n* = 4, 100%), while only 16.3% (*n* = 14) of Italians living in Italy attended online meetings with a therapist, and about 15% of respondents of both populations (Italians living in Italy = 13, and Italians abroad = 11) received psychological support meetings in dual mode. Data were reported in Figure 3.

Almost all of the sample had a computer at home (*n* = 984; 96.1%) with an internet connection (*n* = 980; 95.7%), but, despite this and the increased psychological distress, only 27.1% (*n* = 277) of the respondents showed interest in digital mental health services, while the large majority (44.2%; *n* = 452) did not. Data were reported in Figure 4.

### 3.1. Appropriateness of Psychological Support Services

Analysis showed that 66.6% (*n* = 682) of the respondents evaluated the general appropriateness of psychological support in their country of residence as not adequate (delivered both face-to-face and online), while for 25.2% (*n* = 256) the offer was adequate.

Interestingly, results did not differ meaningfully between Italians living abroad and in Italy; in both cases, the large majority of the sample considered psychological support services to be inappropriate (*n* = 382; 40.9% for Italians living in Italy, *n* = 300; 31.9% for Italian abroad). Similarly, only 9.4% (*n* = 88) of Italians living in Italy, 10.3% (*n* = 97) of those living in other European countries, and 7.8% (*n* = 73) of respondents living in extra-European countries, considered psychological support services adequate.

### 3.2. Utility of Online Psychological Interventions

Most of the general sample considered online psychological interventions very helpful (*n* = 427; 41.7%).

In more detail, the analysis revealed that the majority of Italians both living in Italy and abroad considered online psychological intervention as very helpful (*n* = 222, 23.8%; *n* = 204, 21.6%, respectively), or moderately helpful (*n* = 205, 22.0%; *n* = 203, 21.8 %, respectively). Only a minority of respondents considered online psychological interventions as not useful, and particularly Italians living abroad (*n* = 61, 6.6%) compared to those living in the country (*n* = 39, 4.2%).

While European stakeholders mainly rated online psychological interventions as moderately useful (*n* = 119, 48.0%), the majority of extra-European respondents considered them very helpful (*n* = 112, 50.2%).

Still, 55.5% (*n* = 568) of the total sample considered this method of delivering psychological support as moderately helpful to alleviate distress due to the COVID-19 pandemic and its preventive measures, while only 7.4% (*n* = 76) of the respondents rated the online format as very disadvantageous. The remaining 28.1% (*n* = 288) of the participants labeled online psychological interventions as very advantageous.

### 3.3. Accessibility of Online Psychological Interventions

Nearly half of the sample (*n* = 534, 52.1%) rated online psychological interventions as moderately accessible, while 30.6% (*n* = 313) of the participants considered them as adequately accessible, and only 8.6% (*n* = 88) of the respondents evaluated online interventions as not adequately accessible. Notably, among the latter, 78.1% (*n* = 800) of the respondents had no experience with the services.

When comparing Italians living abroad with inhabitants of Italy, both populations mainly considered online psychological interventions as moderately accessible (Italians abroad: *n* = 253, 27.0%; Italians in Italy: *n* = 281, 30.2%), a minority of the samples considered online psychological services as very accessible (Italians abroad: *n* = 168, 18.0%; Italians in Italy: *n* = 144, 15.4%), while only 4.1% (*n* = 38) of inhabitants of Italy and 5.4% (*n* = 51) of Italians living abroad evaluated the services as not adequately accessible.

### 3.4. Willingness to Adopt Online Psychological Interventions

Responses to the survey revealed that 44.2% (*n* = 453) of the sample was not interested in receiving online psychological interventions, while 27.1% (*n* = 277) of the respondents declared their willingness to be treated digitally, and 20.9% (*n* = 214) were unsure.

Stakeholders from extra-European countries were those more willing to receive an online psychological intervention (*n* = 97, 42.2%), while only 28.5% (*n* = 135) of inhabitants of Italy and 18.8% (*n* = 45) of Italians living in Europe (Italians in Italy excluded) would opt for an online psychological intervention. Complete data were reported in Table 1.

### 3.5. Advantages of Online Psychological Interventions

The primary benefit of the online format has been recognized by 64.5% (*n* = 660) of the total sample in terms of the possibility to fill the gap made up by the geographical distance between the client and the therapist. This was followed by economic reasons (*n* = 398; 38.9%), as online therapy was considered cheaper than face-to-face interventions, and the reduction in waiting lists (*n* = 332; 32.4%), as it guarantees the users immediate access to the treatment. Only 4.3% (*n* = 44) of the respondents identified no advantages of online psychological interventions.

The online psychological intervention has equal or major clinical efficacy compared to face-to-face interventions: 62.1% of Italians in Italy and 31.0% (*n* = 9) of extra-European stakeholders rated this element as an advantage and only 6.9% (*n* = 2) of European stakeholders did so.

### 3.6. Disadvantages of Online Psychological Interventions

Problems with the use of technology were recognized as the main element preventing people from accessing online psychological interventions (*n* = 457; 44.6%).

Low motivation (*n* = 339, 33.1%) and privacy/safety reasons (*n* = 317, 31.0%) were also two disadvantages frequently reported by the stakeholders.

Moreover, the critical and biased attitude of the users (*n* = 309, 30.2%) and the belief that online psychological therapy is less effective than face-to-face interventions (*n* = 294, 28.7%) were listed as other barriers to the use of digital intervention.

Lastly, the idea that online psychological interventions are more expensive than face-to-face interventions represented another potential perceived disadvantage to their usage in 3.9% (*n* = 40) of the respondents. Only 4.3% (*n* = 44) of the sample did not identify any problem with the use of online psychological interventions.

In Italy and extra-European countries, the most voted disadvantage of online psychological interventions was technology issues, identified by 52.5% (*n* = 268) and 40.5% (*n* = 98) of the respondents, respectively. European stakeholders were, indeed, less keen on the use of online therapy as they considered it as having a lower clinical efficacy than face-to-face interventions (*n* = 116, 42.6%).

## 4. Discussion

While there is increasing evidence of the efficacy of online psychological interventions for mental health problems, the implementation of these services into routine practice remains a challenge. These issues became particularly important with the advent of the COVID-19 pandemic, which has forced mental health professionals, and healthcare services to reimagine and redesign their way to work. Involving stakeholders in the planning and implementation of online therapy might increase the spread and success of the interventions, tailored to the specific needs of the population.

Accordingly, this stakeholder survey aimed to reveal valuable insights into the experiences, needs, values, and attitudes towards the use of online interventions for the treatment of mental health disorders. These factors are relevant for the adoption, implementation, and maintenance of such programs, thus increasing the general well-being of the individuals. To our knowledge, this was the first stakeholder survey conducted on this topic and it was specifically aimed at simultaneously investigating the attitude of the Italian population living in the country and abroad, towards psychological support services delivered online. In agreement with other studies, both conducted in Italy [16,39,40] and abroad [41], results from this survey showed that most of the respondents experienced psychological discomfort during the pandemic, with higher percentages among those who contracted the virus.

These results are in line with those reported by Al-Aly et al. (2021) [42] on a large sample of patients who had no mental health diagnoses or treatment for at least two years before becoming infected with COVID-19, and whose experience was compared in the year after they recovered from the infection with that of a similar group of people who did not contract the virus. Results showed that COVID-19 patients were significantly more likely to develop cognitive problems (80%), sleep disorders (41%), depression (39%), stress (38%), anxiety (35%), and opioid use disorder (34%), compared with their counterparts, and that those who need to be hospitalized were at higher risk of developing a mental health problem. Therefore, COVID-19 has a demonstrable marked effect on mental health, and early (digital) treatment of patients facing new or additional mental health challenges during/after COVID-19 can make a crucial difference in reducing the burden of mental health disorders.

However, as shown by this survey, only a few stakeholders asked for psychological support; this may be explained by the fact that, with the advent of the COVID-19 pandemic and related preventive measures, access to psychological services was limited. Indeed, more than half of our respondents considered psychological support services inadequate.

These findings further confirm the need, in the age of COVID-19, to look for an alternative way to provide psychological support due to mobility restrictions. Still, despite the situation at the beginning of the pandemic, some agencies initiated online psychological support services in Italy, such as the SIPES (Società Italiana Psicologia dell’Emergenza) or the SIPO (Servizio Italiano di Psicologia Online); Italians living in Italy favored in-person meetings with the psychotherapist than online therapy. This data probably reflects a lack of knowledge, skepticism, or mistrust towards psychological support interventions delivered online.

This seems particularly true among Italians living in the country, as those living abroad mainly rated online psychological interventions as moderately accessible. These results suggest the need for propaganda and awareness-raising campaigns for Italian citizens, specifically built on their needs and concerns.

Accordingly, research showed that digital therapy has already been successfully established in other countries [43,44,45]. Thus, findings from this survey reveal that online meetings with a therapist were attended mainly by Italians living abroad and that stakeholders from extra-European countries were those more willing to opt for digital therapy, while only a minority of Italian inhabitants of Italy showed interest and inclination in being treated digitally; this is despite most Italians considering online psychological intervention very helpful. This data can be partially explained by the fact that the interviewees listed both advantages and disadvantages related to the use of online psychological interventions. Indeed, online therapy was deemed a good option to overcome the problem of geographical distance, made necessary during the COVID-19-related social isolation [24,46]. However previous studies showed that having to drive long distances and take significant time out of a busy schedule to seek in-person therapy can also be a burden for people in need of help [46], especially those living in rural or remote areas [47,48]. Still, mobility can be a big issue when it comes to individuals with physical limitations accessing mental health care [49]. Moreover, economic reasons were additional acknowledged benefits of online interventions, as it helps to reduce health care costs [50,51]. A systematic review including 50 studies from high-income countries agreed on the cost-effectiveness of online interventions [52].

Further, studies conducted in low-income countries reported that online psychological interventions are particularly advantageous where a few trained mental health providers are present, to obtain both economic and clinical benefits [28,53,54].

While online therapy can potentially be very helpful for people in certain situations, it does not come without some risks and disadvantages over traditional therapy options. Indeed, technological problems can also make it difficult to access treatment, as revealed by respondents to this survey, as well as in other studies [55,56,57]. Notably, most of the stakeholders who identified difficulties in the use of technology as a disadvantage to the use of an online intervention were young adults. This may be because some programs they came into contact with were not very intuitive and usable, as quality issues can affect the way services are provided and received [55,56,57].

Keeping personal information private was another major concern in the use of online treatments. The presence of cameras and microphones can, indeed, lead individuals to be reluctant to open up in full for fear that others may hear [55,58,59,60]. In this regard, online interventions, which are delivered through a secure platform, might receive increased acceptability [61].

### Strengths and Limitations

Different limitations to this study have been identified and listed below:

The online format of the survey limited the sample to those individuals who had access to a computer and the internet. Participants were primarily from Italy, rather than from abroad. This may result in an issue regarding how representative the sample is of the broader population. On the other hand, the fact that data were collected, not only on Italians living in Italy, but also abroad, is an important strength of this study.

Another limitation might be not having included stakeholders other than the general population, or not having employed other methods for data collection rather than the online questionnaire, including focus groups or semi-structured interviews. Asking clinicians, or health facilities would have provided a more exhaustive overview of individuals’ needs and opinions on the use of online psychological support services, and possible solutions to increase their usage.

Furthermore, another limitation might be the absence of information regarding the respondents’ civil status or living conditions, which, during social isolation, had a demonstrable impact on the individuals’ perceived quality of life.

Finally, the characteristics of the population sought do not allow for the socio-economic level of the respondents to be highlighted.

## 5. Conclusions

Considering that viruses know no borders and, increasingly, neither do digital technologies and data, there is an urgent need for the alignment of international strategies for the regulation, evaluation, and use of digital technologies to increase individual mental health. These data help us to gain information on the need and possible barriers to digital therapy, as knowing more about what stakeholders want is the first step toward delivering an effective online psychological intervention

## Figures and Tables

**Figure 1 ijerph-19-07008-f001:**
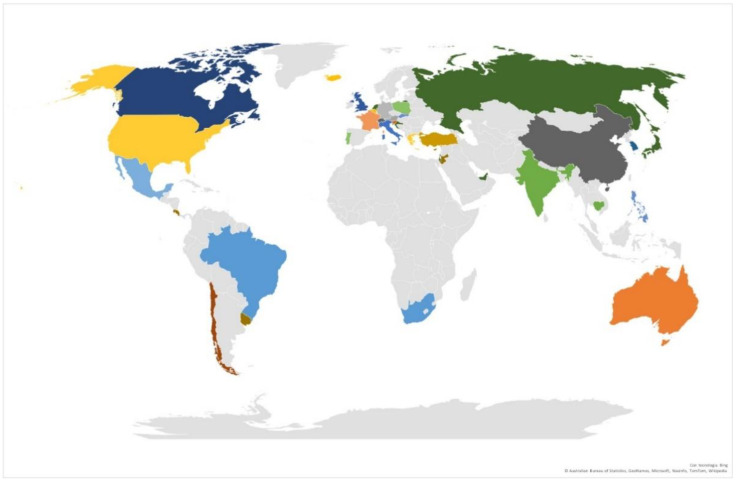
Geographical distribution of the sample. Legend: 0.4%: Russian Federation, United Arab Emirates, The Netherland, Croatia, San Marino, Japan, Israel, Australia; 0.5%: Uruguay, Jordan, Cyprus, Costa Rica; 0.6%: Philippines, Slovakia; 0.7%: Cambodia, India; 0.8%: United Kingdom; 1%: Brazil, South Africa; 1.2%: Greece, USA; 1.3%: Portugal, Poland; 1.4%: Turkey; 1.6%: Chile; 1.7%: Belgium, Iceland; 1.9%: Canada; 2%: France; 2.2%: Austria; 2.5%: Mexico; 2.8%: Germany; 2.9%: Spain; 4.1%: Switzerland, China; 5.4%: South Korea; 8%: Slovenia; and 50%: Italy.

**Figure 2 ijerph-19-07008-f002:**
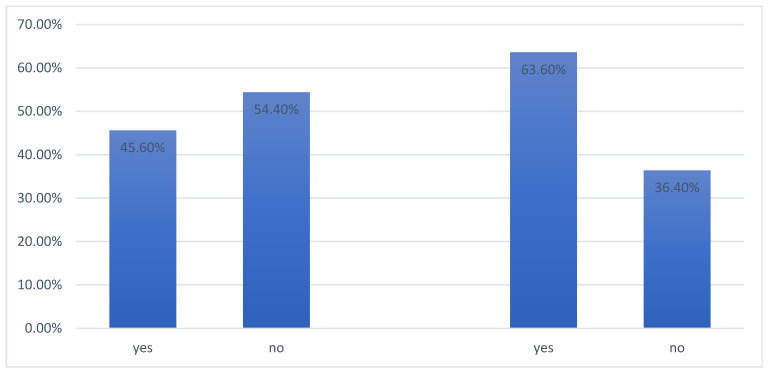
Perceived psychological distress before and during the COVID-19 pandemic.

**Figure 3 ijerph-19-07008-f003:**
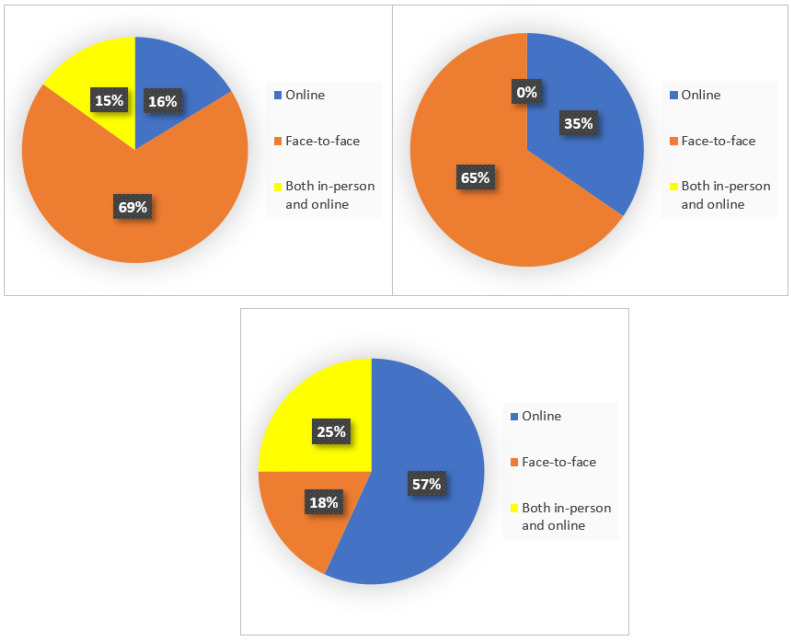
Mode with which psychological support was delivered in Italy, Europe, and extra-European countries.

**Figure 4 ijerph-19-07008-f004:**
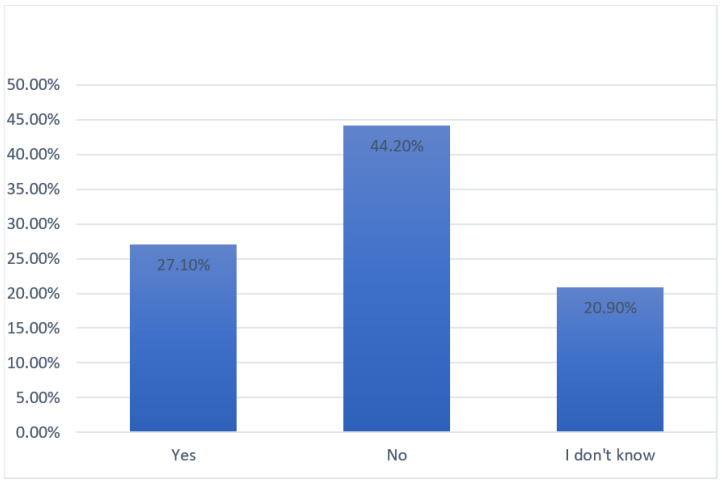
Interested in trying a psychological online intervention.

**Table 1 ijerph-19-07008-t001:** Opinion of Italian stakeholders in Italy, Europe, and extra-European countries about appropriateness, utility, and accessibility of online psychological interventions, and their willingness to use them.

		Italy (*n*, %)	Europe (*n*, %)	Extra-Europe (*n*, %)
Appropriateness				
	Yes	88, 18.7%	97, 39.8%	73, 32.3%
	No	382, 81.3%	147, 60.2%	153, 67.7%
Utility				
	Very useful	221, 47.5%	94, 37.9%	112, 50.2%
	Moderately useful	205, 44.1%	119, 48.0%	84, 37.7%
	Not useful at all	39, 8.4%	35, 14.1%	27, 12.1%
Accessibility				
	Very accessible	143, 31.0%	78, 32.0%	27, 11.7%
	Moderately accessible	281, 60.8%	142, 58.2%	111, 48.3%
	Not accessible at all	38, 8.2%	24, 9.8%	92, 40.0%
Willingness				
	Yes	135, 28.5%	45, 18.8%	97, 42.2%
	No	224, 47.3%	136, 56.7%	93, 40.4%
	Don’t know	115, 24.3%	59, 24.6%	40, 17.4%

## Data Availability

Not applicable.

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
