# Peer review of "Synthesizing Stakeholders Perspectives on Online Psychological Interventions to Improve the Mental Health of the Italian Population during the COVID-19 Pandemic: An Online Survey Study"

_ijerph, 2022, doi:10.3390/ijerph19127008_

Round 1
Reviewer 1 Report
General comment:
The subject is interesting and topical, both from the point of view of the situation related to the COVID 19 epidemic and the use of online intervention modalities.
However, here is an important comment on the general approach and other details on the form.
The same cannot be said for the use of this online survey method. Using a digital survey to study a digital intervention effectively selects the profile of respondents and the quality of the respondents' responses.
Introduction
In particular, the audience of the study is described as including all Italians. The objective of the study should be more precise and focused on people already using digital devices. The use of digital tools excludes a number of them, including the poorest. The characteristics of the population sought do not allow for the socio-economic level of the respondents to be highlighted. The choice contributes to erase the effect of social inequalities in health, which plays a real role in mental health problems and also in the use of this type of digital tool.
Materials and methods
Line 141: Where do the 9 possible options come from? What are the references or construction methods?
Results
Line 179: Map color legend ?
Line 193: Table 1 would be better placed in the Appendix
Line 201 : Figure 2: put the yes together and the no together
Line 211: Attention to editing: move "n=4" for South Korea and US
Line 219 : Figure 3: Titles of the graphs are missing (Italy, Europe, extra-european)
Interest of the graph ? why not show the difference between before and during the pandemic ?
Strengths and limitations
One of the main limitations is not mentioned: the failure to take into account the socio-economic level. If no particular attention is paid to the psychological interventions, they will be a factor in the aggravation of social inequalities in health.
Author Response
Introduction
In particular, the audience of the study is described as including all Italians. The objective of the study should be more precise and focused on people already using digital devices. The use of digital tools excludes a number of them, including the poorest. The characteristics of the population sought do not allow for the socio-economic level of the respondents to be highlighted. The choice contributes to erase the effect of social inequalities in health, which plays a real role in mental health problems and also in the use of this type of digital tool.
Thanks for your suggestions; to our knowledge, the use of technology is not transversal to the socio-cultural characteristics of the population (Hirschman, R.S. 1982; Combi, 2016; Rakhimova et al., 2017). We preferred to focus on the problem that age could impact results due to digital literacy, which however did not occur in the results in fact a good group of people over 50 responded.
Materials and methods
Line 141: Where do the 9 possible options come from? What are the references or construction methods? We relied on a study from the literature. Now you find the bibliographic reference in the text.
Results
Line 179: Map color legend?
Thanks for the observation, but we believe it is not necessary to make a legend on the colors: the States that participated are superior to the possible colors, we also believe that everyone possesses the minimum geographical knowledge. The legend as a list of participating States seems to us sufficient.
Line 193: Table 1 would be better placed in the Appendix
Thanks for the tip, Table 1 moved to the appendix
Line 201 : Figure 2: put the yes together and the no together
Thanks for the observation, but from our point of view it makes no sense to put the yes and the no together because we are talking about two distinct moments to make the comparison, that is before and during the pandemic.
Line 211: Attention to editing: move "n=4" for South Korea and US
Thanks for your correction
Line 219 : Figure 3: Titles of the graphs are missing (Italy, Europe, extra-european)
Interest of the graph ? why not show the difference between before and during the pandemic?
The title is present. This distinction is fundamental to understand what are the differences around the world regarding the way in which psychological support services are provided. The distinction between before and during the pandemic is a good starting point but we do not have the data at the moment to be able to do this in depth.
Strengths and limitations
One of the main limitations is not mentioned: the failure to take into account the socio-economic level. If no particular attention is paid to the psychological interventions, they will be a factor in the aggravation of social inequalities in health.
Thanks for your suggestion: we have inserted this limit in the corresponding section of the article (line 410-411).
Reviewer 2 Report
First, I would like to congratulate the authors for the excellent work presented. The topic is very current and relevant.
There is a good theoretical basis, and the objectives and methods are very clear. The results are carefully described and the discussion is well-founded.
I have a few comments about the article:
1. I would like to suggest that the title be more objective. While it does show the purpose of the study, making it more concise will be better for the reader.
2. When reading the results, especially regarding the disadvantages of online therapy, I was wondering if the therapy time, the number of times per week or month, and also the professional's therapeutic line could also affect the client's disposition. Did the authors take these variables into account? Could they also interfere?
Author Response
I have a few comments about the article:
1. I would like to suggest that the title be more objective. While it does show the purpose of the study, making it more concise will be better for the reader.
Thanks for the suggestion: in reality we believe that the title allows the reader to better understand what he will find in the text and in this case specifying what the work will be done towards the stakeholders seems fundamental to us. For this reason we keep the title that was previously chosen.
When reading the results, especially regarding the disadvantages of online therapy, I was wondering if the therapy time, the number of times per week or month, and also the professional's therapeutic line could also affect the client's disposition. Did the authors take these variables into account? Could they also interfere?
We also thank you for this observation: these variables are absolutely important but to have this information we would have had to ask further questions in the survey that would have been so too long and more exposed to the risk of the drop. We therefore preferred to reduce the number of questions to get a first overview on the subject. Future studies will undoubtedly focus on this fundamental perspective when researching purely clinical topics.
Round 2
Reviewer 1 Report
Thank you for the clarification and especially the addition of the fundamental limit of the socio-economic status of users in the use of digital tools.